# Sensing intracellular calcium ions using a manganese-based MRI contrast agent

Ali Barandov[1], Benjamin B. Bartelle[1], Catherine G. Williamson[1], Emily S. Loucks[1], Stephen J. Lippard[2] & Alan Jasanoff[1,3,4]

Calcium ions are essential to signal transduction in virtually all cells, where they coordinate processes ranging from embryogenesis to neural function. Although optical probes for intracellular calcium imaging have been available for decades, the development of probes for noninvasive detection of intracellular calcium signaling in deep tissue and intact organisms remains a challenge. To address this problem, we synthesized a manganese-based para-magnetic contrast agent, ManICS1-AM, designed to permeate cells, undergo esterase cleavage, and allow intracellular calcium levels to be monitored by magnetic resonance imaging (MRI). Cells loaded with ManICS1-AM show changes in MRI contrast when stimulated with pharmacological agents or optogenetic tools; responses directly parallel the signals obtained using fluorescent calcium indicators. Introduction of ManICS1-AM into rodent brains furthermore permits MRI-based measurement of neural activation in optically inaccessible brain regions. These results thus validate ManICS1-AM as a calcium sensor compatible with the extensive penetration depth and field of view afforded by MRI.

[1] Department of Biological Engineering, Massachusetts Institute of Technology, 77 Massachusetts Ave. Rm. 16-561, Cambridge, MA 02139, USA. [2] Department of Chemistry, Massachusetts Institute of Technology, 77 Massachusetts Ave. Rm. 16-561, Cambridge, MA 02139, USA. [3] Department of Brain & Cognitive Sciences, Massachusetts Institute of Technology, 77 Massachusetts Ave. Rm. 16-561, Cambridge, MA 02139, USA. [4] Department of Nuclear Science & Engineering, Massachusetts Institute of Technology, 77 Massachusetts Ave. Rm. 16-561, Cambridge, MA 02139, USA. These authors contributed equally: Ali Barandov, Benjamin B. Bartelle. Correspondence and requests for materials should be addressed to A.J. (email: jasanoff@mit.edu)

Calcium imaging techniques are among the most widely used experimental methods in modern biology, but the technology for measuring large-scale calcium signaling dynamics noninvasively remains limited. With optical calcium reporters it is now possible to perform functional imaging of intracellular $[Ca^{2+}]$ at depths of about a millimeter intact tissue[1], but for most vertebrate species this only gives access to a small fraction of the volumes of experimental interest. Implantable endoscopes and prisms permit measurements in deeper structures, but only over limited fields of view[2,3]. Hybrid techniques like photoacoustic tomography achieve submillimeter imaging resolution with considerably greater tissue penetration than conventional optics. Although suitable calcium imaging probes have been developed[4,5], their application is limited by trade-offs between depth and resolution[6], and invasive surgery remains a requirement for these approaches. In order to measure calcium signaling processes in tissue regions of arbitrary size and depth, there is therefore an urgent need for probes that are compatible with truly noninvasive imaging modalities.

Magnetic resonance imaging (MRI) is a uniquely powerful neuroimaging technique that could provide a basis for wide-field deep-tissue calcium imaging in animals and humans[7]. MRI achieves a combination of unlimited depth penetration, relatively high 3D spatial resolution (<100 μm in some contexts), and sensitivity to a wide variety of contrast mechanisms. There have been extensive efforts to design responsive MRI contrast agents for monitoring analytes such as metal ions and neurotransmitters[8–11]. To target $Ca^{2+}$, most progress has been made using probes based on gadolinium complexes or superparamagnetic iron oxide nanoparticles[12–14]. These sensors transduce varying calcium concentrations into changes in longitudinal ($T_1$) or transverse ($T_2$) relaxation rates, which may be visualized, respectively, by $T_1$- or $T_2$-weighted MRI. Although extracellular calcium changes have been successfully measured in vivo using such probes[15,16], accessing intracellular calcium ions has been more challenging, owing in part to the difficulty of delivering existing polar or bulky MRI calcium sensor architectures into cells.

We speculated that a successful intracellular calcium sensor for MRI could be produced by following design principles used in constructing fluorescent calcium imaging dyes like Fura-2, X-Rhod-1, and Oregon Green BAPTA[17]. These indicators each consist of a planar cell-permeable aromatic fluorophore functionalized with a selective calcium binding moiety derived from 1,2-bis(2-aminophenoxy)ethane-N,N,N′,N′-tetraacetic acid (BAPTA). Although BAPTA is hydrophilic and membrane-impermeant, it may be modified with acetomethoxy (AM) ester groups to produce a neutral, inactive compound that is readily internalized into cells[18,19]. After internalization, the AM esters undergo hydrolysis catalyzed by intracellular esterases. This process traps the probes within the cell cytosol and restores their ability to sense calcium.

We recently created a family of cell-permeable phenylenediamido (PDA)-based $Mn^{3+}$ complexes that act as $T_1$ MRI contrast agents capable of selective cell labeling[20]. We reasoned that conjugation of such complexes to BAPTA-based chelators would result in candidate MRI calcium sensors with similar physicochemical properties to optical calcium imaging dyes. Such probes could be administered to cells in an AM ester-modified form, which would be internalized and activated (Fig. 1a). Interactions between the BAPTA and Mn-PDA groups of the activated sensor could then provide a basis for coupling intracellular calcium concentration changes to $T_1$-weighted MRI signals.

## Results

### Building blocks for intracellular MRI calcium sensors.
To examine the potential of Mn-PDA contrast agents to function as building blocks for our sensor design, we performed MRI measurements of BAPTA interactions with three previously introduced paramagnetic chelate complexes[20] (Fig. 1a inset). Addition of 1.1 equiv of BAPTA to two of the Mn-PDA complexes, Mn**L1** and Mn**L3**, produces notable effects on the $T_1$-weighted MRI contrast induced by these compounds in buffer; a third compound, Mn**L2**, is unresponsive to BAPTA. Importantly, these BAPTA-dependent contrast changes are reversed by addition of equimolar $Ca^{2+}$ (Fig. 1b). This result is consistent with the possibility that the Mn-PDA moieties compete with $Ca^{2+}$ for binding to BAPTA molecules. The changes could be quantified in terms of $T_1$ relaxivity ($r_1$) values, which are defined as the slope of the $T_1$ relaxation rate ($1/T_1 = R_1$) vs. concentration of each paramagnetic complex. Calcium-dependent changes in $r_1$ observed for Mn**L1** and Mn**L3** in the presence of BAPTA are 105% and 23%, respectively; both are highly significant (t-test $p < 7 \times 10^{-6}$).

Although mixtures of Mn**L1** or Mn**L3** with BAPTA both showed promising calcium-dependent MRI properties, we were concerned that competition between the PDA and BAPTA chelators for access to manganese might compromise stability of the Mn-PDA complexes while promoting the $r_1$ changes we observed. To test for this possibility, we examined the spectroscopic behavior of Mn**L1** and Mn**L3** in the presence of BAPTA over time, and found indeed that the Mn**L1**-containing mixture displays sharp changes indicative of degradation, whereas the Mn**L3** mixture remains unperturbed and apparently stable (Fig. 1d). These results were confirmed by high resolution mass spectrometry of the two mixtures. Even after 24 h, the Mn**L3** mixture with BAPTA displays a prominent base peak associated with [Mn**L3** + Cl]⁻ ($m/z = 419.77$) but no peak for apo **L3**. In contrast, the Mn**L1** mixture with BAPTA develops a strong peak for apo **L1** ($m/z = 347.37$), indicating dissociation of the Mn**L1** complex (Supplementary Figure 1). These results together suggest that Mn**L3** and BAPTA constitute a suitable pair of building blocks for intracellular calcium sensor construction, but that Mn**L1** is insufficiently stable despite its greater calcium-dependent $r_1$ changes in the presence of BAPTA.

### Synthesis and characterization of ManICS1 and ManICS1-AM.
To form our first manganese-based intracellular calcium sensor (ManICS1) from Mn**L3** and BAPTA, we synthesized ManICS1 (**14**) and its AM ester derivative ManICS1-AM (**16**) using a series of multistep reactions starting from 5-methyl-2-nitrophenol (**1**) (Fig. 2, Supplementary Figure 2, and Supplementary Methods). Compound **7**, as the parent BAPTA derivative for preparing ManICS1, was synthesized as previously reported[21]. Conversion of the aldehyde group of this compound to a carboxylic acid was performed through Pinnick oxidation[22], which enables high yields without requirement for transition metal catalysis. Oxidation of aldehyde using milder conditions was unsuccessful owing to the presence of the electron-donating tertiary amine groups. Successive coupling of a polyethylene glycol linker and the bifunctional **L3** derivative, **L3COOH** (**12**), to compound **7** was achieved using benzotriazol-1-yl-oxytripyrrolidinophosphonium hexafluorophosphate, which produced higher yields than other coupling reagents. The resulting compound **15** was metallated to form ManICS1-AM by reaction with $Mn(OAc)_3 \cdot 2(H_2O)$ in acetonitrile in the presence of N,N-diisopropylethylamine. Enzymatic hydrolysis of ManICS1-AM or direct coupling of compound **11** to Mn**L3COOH** (**13**) afforded ManICS1 in moderate yields (Fig. 2).

We measured the calcium responsiveness of ManICS1 in buffer by titrating the compound with calcium chloride over a concentration range relevant to intracellular signaling (0–100 μM). Images indicate clear enhancement of $T_1$-weighted MRI intensity

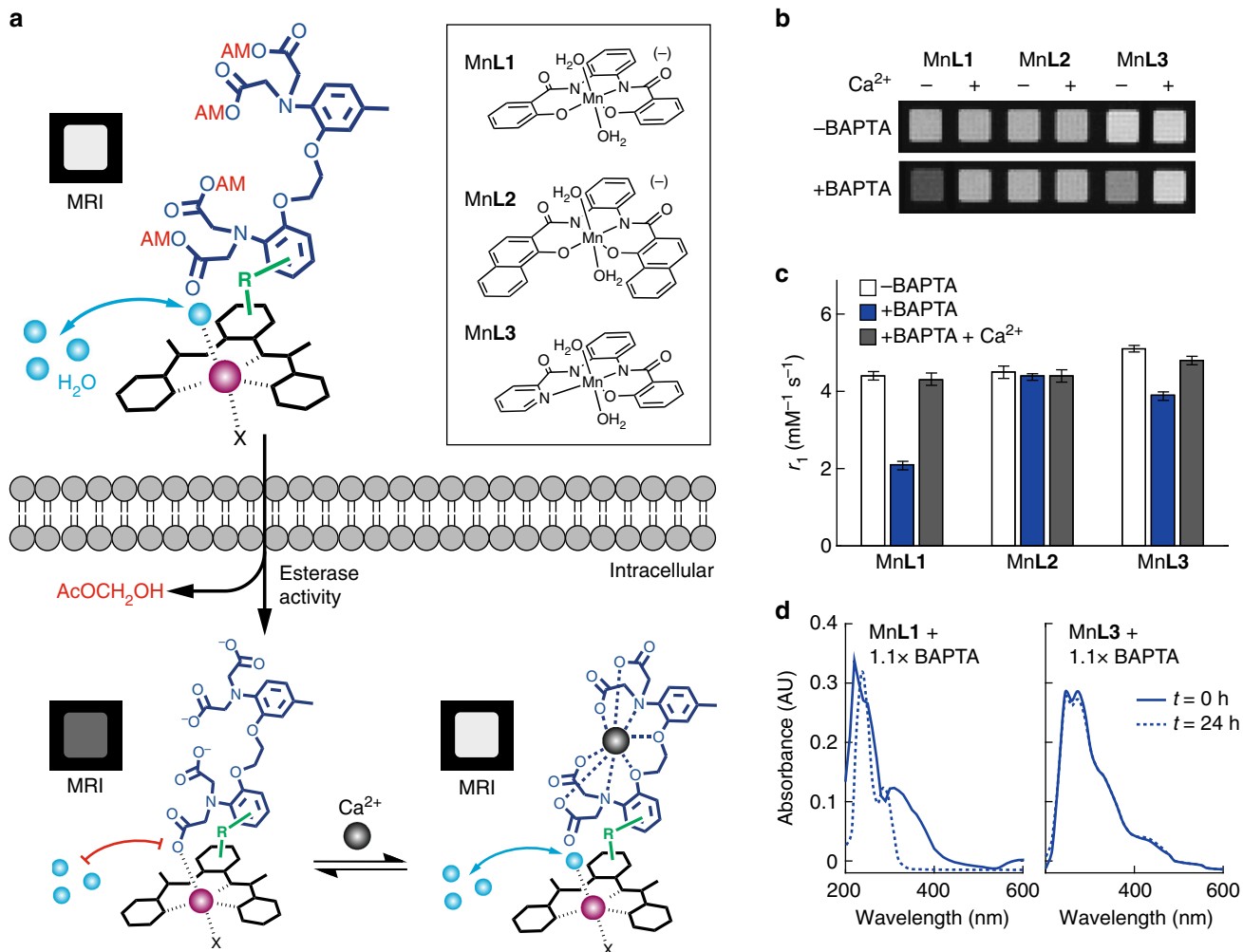

**Fig. 1** Design of cell permeable sensors for calcium-dependent molecular fMRI. **a** The design consists of a cell permeable paramagnetic platform (black complex, Mn-PDA candidate complexes shown at top right), a BAPTA-based calcium chelator (dark blue), and a linker connecting them (green, R). Prior to cell entry (top) the BAPTA carboxylates are protected with cleavable AM esters (red), and water exchange (cyan spheres) is expected to take place at the paramagnetic metal center (purple), leading to $T_1$-weighted MRI signal enhancement (labeled MRI). When the agent enters cells (bottom left), the AM esters are cleaved, liberating the sensor in its calcium-free "off" state; in this state, water exchange is expected to be blocked by interactions between the BAPTA moiety and $Mn^{3+}$, leading to low MRI signal. When calcium binds (bottom right), the MRI signal could increase again as interactions between the BAPTA and paramagnetic platform are reduced. **b** Evaluation of interactions between untethered BAPTA and Mn-PDA contrast agents. In the absence of BAPTA, the $Mn^{3+}$ complexes (40 μM) do not produce calcium-dependent $T_1$-weighted MRI contrast changes (top), but in the presence of 1.1 equiv BAPTA, both MnL1 and MnL3 display sensitivity to addition of 1 mM calcium (bottom). **c** Longitudinal relaxivity ($r_1$) values corresponding to the conditions in panel **b**. Error bars denote SD of four measurements. **d** Optical spectra of contrast agents MnL1 and MnL3 in the presence of BAPTA after incubation for 0 h (solid) or 24 h (dashed), indicating comparative stability of MnL3

as calcium ions are added (Fig. 3a). No image changes are observed when $Mg^{2+}$ is added in place of $Ca^{2+}$, indicating specificity of the responses to calcium. Apparent relaxivity values were determined for each condition (Fig. 3b). The data indicate that calcium binding induces a 34% increase in the relaxivity of ManICS1, with $r_1$ values ranging from $3.6 \pm 0.1$ to $5.1 \pm 0.1$ $mM^{-1}s^{-1}$ over the full calcium range (unless otherwise noted, all error margins indicate SEM of three or more independent measurements). As expected, no significant $r_1$ change is observed with $Mg^{2+}$ titration. The dissociation constant for calcium binding to ManICS1 was determined to be $18 \pm 12$ μM (error margin from curve fit). This $K_d$ value is higher than most high affinity fluorescent indicators used for intracellular $Ca^{2+}$ imaging, but is still sufficient for converting $[Ca^{2+}]$ fluctuations by as little as 1 μM from baseline into $T_1$-weighted image changes of ~1% or higher, equivalent to signals commonly detected in functional MRI experiments.

Although we suspected that the dominant interactions between BAPTA and $Mn^{3+}$ would take place within individual ManICS molecules, there was also a possibility that intermolecular coordination or hydrophobic stacking interactions could contribute to the relaxivity effects we saw. To test for such phenomena, we performed separate calcium titration series with three different concentrations of ManICS1 (Supplementary Figure 3). We observed weak concentration-dependent differences in the titration curves, but no significant discrepancies among $K_d$ values obtained at different concentrations (t-test $p \geq 0.26$), arguing that intermolecular interactions between ManICS monomers do not play a substantial role in their calcium ion sensitivity at concentrations relevant to molecular imaging.

As an additional test of our proposed molecular mechanism for calcium sensing by ManICS1, we formed a manganese-free ManICS1 analog and measured its calcium affinity by titration with a spectroscopic readout (Supplementary Figure 4). The $K_d$ of

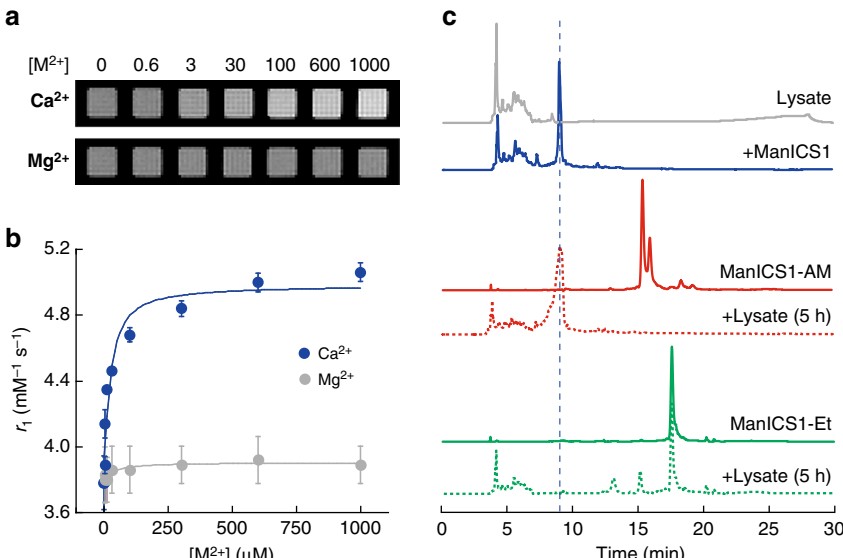

**Fig. 2** Synthesis of ManICS1 and ManICS1-AM. *i*) AcOCH₂Br, DIEA (67%); *ii*) NaH₂PO₄, NaClO₂ (95% yield); *iii*) PyBOP, DIEA, NH₂-PEG₄-NHBoc (63%); *iv*) TFA, DCM (98%); *v*) NaOH (6 eq), H₂O (77%); *vi*) Mn**L3COOH** (**13**), HATU, DIEA (33%); *vii*) **L3COOH** (**12**), PyBOP, DIEA (67%); *viii*) Mn(OAc)₃.2(H₂O) (77%); *ix*) 10% (v/v) cell lysate, MOPS (25 mM, pH 7.9). Additional details described in Supplementary Methods

**Fig. 3** ManICS1 reports calcium-dependent MRI signal changes in buffer and MnICS1-AM undergoes enzymatic hydrolysis. **a** $T_1$-weighted images of 40 μM ManICS1 with addition of varying micromolar concentrations of CaCl₂ (top) or MgCl₂ (bottom) in MOPS buffer, pH 7.4. **b** Relaxivity changes corresponding to conditions in **a**. Ligand-depleting bimolecular binding models were used to generate the fitted curves shown. Error bars denote SD of four measurements. **c** Reversed phase HPLC traces of 10% (v/v) HEK293 cell lysate (gray), ManICS1(blue), ManICS1-AM (solid red), ManICS1-AM treated with cell lysate for 5 h (dotted red), ManICS1-Et (solid green), ManICS1-Et treated with cell lysate for 5 h (dotted green). Vertical dashed line indicates expected elution time of ManICS1

this compound is only 200 nM, similar to the calcium affinity of free BAPTA. The substantially higher $K_d$ of full ManICS1 compared with this value is consistent with the model (Fig. 1a), in which calcium binding by the BAPTA moiety must compete with intramolecular manganese binding, incurring loss of apparent calcium affinity.

Yet further support for the model (Fig. 1a) comes from studies of ManICS1-AM. The AM ester-derivatized compound shows a

relaxivity of $5.5 \pm 0.2$ mM$^{-1}$ s$^{-1}$, about 45% higher than ManICS1 and within error of the relaxivity of the parent complex MnL3 ($5.2 \pm 0.1$ mM$^{-1}$ s$^{-1}$). This is consistent with the likelihood that uncharged ester moieties of ManICS1-AM interact less avidly with the Mn$^{3+}$ ion, and thus do not restrict water access or affect the complex's relaxivity as much as the corresponding acidic moieties in ManICS1 itself. We found that ManICS1-AM is resistant to transmetallation, indicating that replacement of the chelated manganese ion is not a confound in our results (Supplementary Figure 5).

To demonstrate that the ester groups of ManICS1-AM are capable of undergoing cleavage in the cytosolic milieu, we incubated this compound in clarified cell lysate (10% by volume) and compared the high performance liquid chromatography (HPLC) time course to that of ManICS1 and ManICS1-AM incubated in buffer alone. We found that five hours of exposure to cell lysate is sufficient to result in complete conversion of the ManICS1-AM HPLC peak into a product that elutes at the same time as ManICS1 (Fig. 3c). The identity of this product as ManICS1 is confirmed by mass spectrometry, indicating that all four esters of ManICS1-AM are released within the incubation time. As a further test of the behavior of ManICS1 ester derivatives, we synthesized ManICS1-Et, in which all four BAPTA carboxylates are modified by ethyl esters rather than AM groups. HPLC data indicate that ManICS1-Et incubation does not yield ManICS1 after five hour incubation with lysate; this offers the possibility of achieving genetically targeted cleavage and intracellular accumulation of ManICS1-Et by ectopic expression of appropriately selective esterases in future work[23].

**ManICS1-AM enables intracellular calcium-sensitive MRI.** To test the ability of ManICS1-AM to enable readouts of intracellular calcium by $T_1$-weighted MRI, we began by examining its propensity to accumulate within cells. We incubated cultured HEK293 cells with 10 μM ManICS1-AM or ManICS1 for 30 min, followed by washing and MRI analysis (Fig. 4a). We found that cells incubated with the cell permeable ManICS1-AM undergo a substantial increase in $R_1$ that persists above basal levels for up to 24 h, while cells labeled with ManICS1 experience a somewhat lesser increase in $R_1$ that returns to baseline within 5 h. This is consistent with the hypothesis that ManICS1-AM is internalized and converted to ManICS within cells (Fig. 1a), resulting in intracellular trapping of the polar esterase product and extended retention with respect to ManICS introduced from outside cells. To assess the subcellular localization of manganese following ManICS1-AM or ManICS1 labeling, we isolated cytosolic fractions from cell lysates and analyzed the samples by inductively coupled plasma mass spectrometry (ICP-MS). The ICP-MS results indicate that only ManICS1-AM labeling produces cytosolic elevations in manganese content (Fig. 4a inset and Supplementary Figure 6). These results are again consistent with the hypothesis that ManICS1-AM penetrates cells and is retained in the cytosol. Once internalized, the contrast agent moreover appears to be well-tolerated. Tests for acute toxicity and long-term viability of HEK293 cells incubated with ManICS1-AM show no evidence of ill-health with respect to controls (Supplementary Figure 7).

To assess the calcium-dependent contrast properties conferred by ManICS1 inside cells, we titrated intracellular calcium levels using a procedure previously applied to test fluorescent calcium

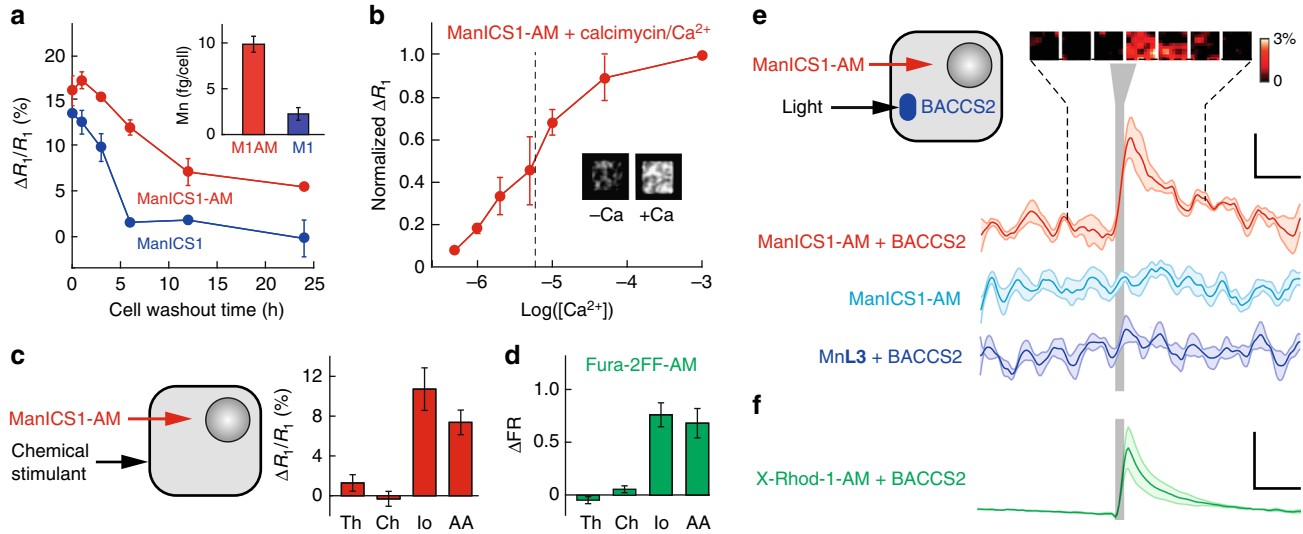

**Fig. 4** ManICS1 reports calcium-dependent MRI signal changes in cells. **a** Washout time course of $\Delta R_1/R_1$ vs. time for HEK293 cells preincubated with 10 μM ManICS1-AM (red) or ManICS1 acid form (blue). Elemental analysis of cytosolic fractions from incubated cell lysates (inset) indicates intracellular manganese accumulation in ManICS1-AM (M1AM)-treated cells but not ManICS1 (M1)-treated cells. **b** Titration of extracellular calcium concentration equilibrated in the presence of 10 μM of the Ca$^{2+}$ ionophore calcimycin in ManICS1-AM-loaded cells. A midpoint of calcium-induced changes occurs at [Ca$^{2+}$] = 5 μM. Inset compares MRI scans of cell pellets imaged in the absence (left) vs. presence (right) of 1 mM calcium. **c** Calcium responses measured from HEK293 cells labeled with ManICS1-AM and treated by extracellular addition of chemical stimulants (left). Significant $R_1$ changes are observed in response to ionomycin (Io) and arachidonic acid (AA) ($p \leq 0.001$), but not thapsigargin (Th) or carbachol (Ch) ($p \geq 0.2$). **d** Responses measured by fluorescence spectroscopy from cells loaded with the fluorescent calcium indicator Fura-2FF-AM under stimulation conditions as in **c**. **e** Cells were loaded by incubation with 40 μM ManICS1-AM, transfected with the light sensitive Orai calcium channel activator BACCS2, and stimulated with 480 nm light (diagram at left). The red time course shows resulting changes in $T_1$-weighted signal as a function of time, before, during, and after stimulation (vertical gray bar). Inset at top depicts image snapshots binned over successive 240 s windows during the time series portion indicated by dashed lines and indicating percent signal changes observed at a voxel level. Stimulus-dependent signal changes were not observed in analogous experiments performed using cells lacking BACCS2 (cyan) or cells labeled with MnL3 instead of ManICS1-AM (blue). **f** Fluorescence time course observed from BACCS2-expressing cells incubated with 5 μM X-Rhod-1-AM and stimulated as in panel **e**. Scale bars: horizontal = 300 s, vertical in **e** = 0.3%, **f** = 20%. Error bars represent SEMs of **a** three, **b** three, **c** six, and **d** six independent measurements. Shading in **e** and **f** represents SEM from five measurements

indicators[24], and examined the profile of resulting $R_1$ changes. HEK293 cells were labeled with 10 μM ManICS1-AM and then equilibrated with varying buffered calcium levels in the presence of the calcium ionophore calcimycin (10 μM)[25]. Results demonstrate progressive changes in $R_1$ as $[Ca^{2+}]_i$ ranges from 1 to 100 μM, establishing a midpoint ($EC_{50}$) for intracellular calcium sensing by ManICS1 of about 5 μM (Fig. 4b). This value shows that ManICS1 is more sensitive to calcium in the intracellular milieu than in the buffer conditions, perhaps due to interactions between the contrast agent and macromolecular solutes that could perturb the sensor's conformational equilibria or coordination sphere.

To assess the potential for internalized ManICS1 to report stimulus-induced cytosolic calcium concentrations as $R_1$ changes detectable by MRI, we again labeled HEK293 cells with ManICS1-AM, then challenged the cells with pharmacological agents that elevate cytosolic calcium levels (Fig. 4c). Ionomycin and arachidonic acid[24,26] both produce 8–10% increases in mean $R_1$ values recorded within 20 min of stimulation. Cells treated with thapsigargin or carbachol, which are thought to cause only short-lived $Ca^{2+}$ responses[27], do not show an elevated $R_1$. Control measurements using the fluorescent calcium indicator derivative Fura-2FF-AM collected using similar stimulus conditions closely parallel the results obtained by MRI of ManICS1-AM-labeled cells (Fig. 4d), showing that the MRI results provide an accurate measure of intracellular calcium perturbations.

To examine the reversibility and dynamics of ManICS1-based calcium sensing we used the optogenetic $Ca^{2+}$ actuator BACCS2[28] to stimulate contrast agent-labeled cells while performing functional imaging with MRI. In preparation for these experiments, cells were embedded in a gel matrix that permitted effective $Ca^{2+}$ exchange at high cell density. BACCS2-expressing cells loaded with ManICS1-AM show dynamic light-dependent image changes averaging 0.8 ± 0.2% in amplitude (Fig. 4e), matching a time course obtained using optical measurements of cells loaded with the fluorescent calcium indicator X-Rhod-1[15] (Fig. 4f). Stimulation of ManICS1-AM-labeled cells that do not express BACCS2 or MnL3-labeled cells expressing BACCS2 produces negligible MRI effects. Both are significantly lower than those observed with ManICS1-AM in BACCS2-expressing cells (t-test $p \leq 10^{-5}$), revealing the calcium specificity of $T_1$-weighted imaging signals obtained with ManICS1-AM inside living cells.

**ManICS1 detects deep brain activation in rats.** To determine whether ManICS1-AM could enable detection of intracellular calcium signals in vivo, we injected the probe into the brains of adult rats and examined responses to stimulation with potassium ions, which induce neural depolarization and calcium concentration changes in brain tissue[29,30]. Intracranial infusion of ManICS1-AM into the striatum results in substantial $T_1$-weighted MRI signal enhancement over a ~4 mm diameter region around the injection site, as well as in more remote tissue along the ventricles (Fig. 5a, b). The contrast enhancement persists for over 90 min without significant loss of signal (t-test $p = 0.42$, $n = 4$)

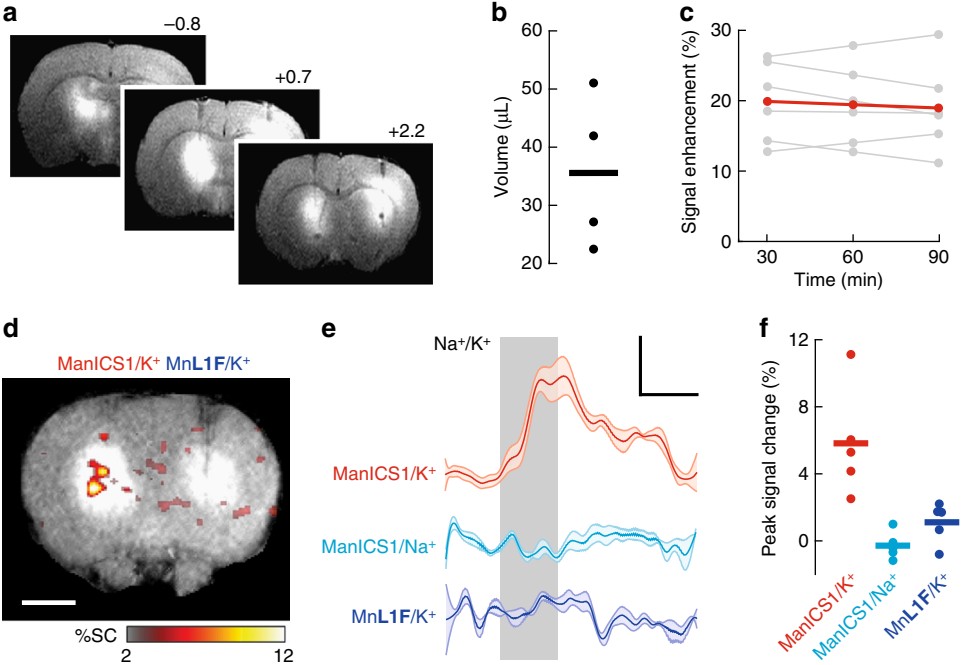

**Fig. 5** ManICS1-AM enables detection of neural activation in rat brain. **a** $T_1$-weighted MRI showing broad contrast enhancement following infusion of 15 μL ManICS-1 (left) or calcium-insensitive control agent (Mn**L1F**, right) into rat striatum. Rostrocaudal coordinates with respect to bregma indicated. **b** Estimated volume of signal enhancements within 10% of peak values after ManICS1-AM infusion in four animals (mean = 36 ± 7 μL). **c** Plot of peak MRI signal in ManICS1-AM infused brain regions over time shows a mean signal enhancement of 20 ± 2% with an average of 0.9% signal decrease per hour with respect to image intensity in unenhanced tissue (red); the signal decrease from 30 to 90 min post infusion was not statistically significant (t-test $p = 0.42$, $n = 6$). Data from individual animals shown in gray (scale bar = 3 mm). **d** 1 μL $K^+$ infusion causes $T_1$-weighted MRI signal increases in the presence of predelivered ManICS1-AM (left) but not Mn**L1F** (right). Average peak signal change across multiple animals ($n = 5$) is indicated by the color scale superimposed on a high resolution $T_1$-weighted image of a representative rat. Scale bar = 3 mm. **e** Region of interest analysis shows the time course of signal changes observed during $K^+$ or $Na^+$ in the presence of ManICS1 (red and cyan, respectively), and during $K^+$ stimulation (vertical gray bar) in the presence of calcium-insensitive Mn**L1F** (blue). Shading represents SEM of five measurements. **f** MRI signal changes observed in individual animals within one minute of $K^+$ or $Na^+$ treatment offset for the conditions in **e**. Mean signal changes observed during potassium stimulation in the presence of ManICS1-AM was significantly greater than results from both controls (t-test $p \leq 0.016$, $n = 5$)

(Fig. 5c), differing markedly from the behavior of hydrophilic MRI contrast agents that do not enter cells and that typically clear from the rodent brain within two hours[31]. The mean $T_1$-weighted signal enhancement of ~20% we observe due to ManICS infusion corresponds to a calcium-free concentration of ~19 μM in tissue, assuming equivalent $r_1$ values in vivo and in vitro.

Infusion of artificial cerebrospinal fluid (aCSF) formulated isotonically with 125 mM KCl elicits a robust signal change proximal to the infusion site in ManICS1-AM-infused brain areas (Fig. 5d). The ManICS1-dependent signal rises quickly, with an average signal change of $5.8 \pm 1.2\%$ at stimulus offset that subsides slowly to baseline after the KCl infusion stops (Fig. 5e). KCl injection in the presence of the calcium-insensitive contrast agent MnL1F—a close variant of the soluble cell-permeable MnL1 chelate (see Supplementary Methods)—or infusion of standard aCSF containing 125 mM NaCl in the presence of ManICS1-AM elicits negligible mean responses of $-0.2 \pm 0.8\%$ and $1.2 \pm 1.2\%$, respectively, at stimulus offset with respect to baseline. The signal change observed under the ManICS1-AM K$^+$ stimulation condition differs significantly from signals observed under both control conditions ($t$-test $p \leq 0.016$, $n = 5$) (Fig. 5f), consistent with the expected calcium-sensing mechanism of ManICS1 and with results obtained in cells.

## Discussion

Our experiments thus demonstrate a cell-permeable manganese-based MRI contrast agent that emulates properties of fluorescent probes for cytosolic Ca$^{2+}$ imaging, and that can detect signaling events in deep tissue. The ManICS1 calcium sensor introduced here incorporates membrane-permeable building blocks and can exploit the AM ester-based approach for cell labeling and cytosolic trapping of the probe. Upon internalization, ManICS1-AM reports calcium levels consistent with readouts from optical calcium sensors and compatible with $T_1$-weighted functional MRI in rat brain, suggesting that the new probe could be used for spatiotemporal mapping of calcium signaling processes previously accessible only to optical imaging approaches, but with the expanded depth and field of view afforded by MRI.

Further development of this class of MRI probe design could include chemical modifications to the manganese-binding moiety, the calcium-specific chelator, and the linker between them, with the aims of increasing calcium affinity and optimizing interactions between the calcium chelator and the paramagnetic complex. Enhanced calcium binding, for instance, might be conferred by addition of more electron-donating groups to the phenyl groups of the BAPTA subunit. Such modifications would in turn increase the dynamic range of relaxivity changes and sensitivity to low levels of calcium. Delivery strategies may also need to be refined for organ or organismic-scale applications, if properties of ManICS1-AM itself do not yield desirable tissue distributions. In the brain, for instance, methods for delivery past the blood–brain barrier might be required.[7] Manipulating the esterase sensitivity of ManICS derivatives might also provide a means for tuning tissue localization properties.[20,23] The current probe, however, already enables measurement of responses to multiple biologically relevant stimuli in accessible tissue regions, demonstrating its immediate utility for a variety of applications in basic biology and biomedicine.

## Methods

**Synthesis**. All information regarding the synthesis and characterization of Man-ICS1, ManICS1-AM, and related compounds is reported in the Supplementary Methods section.

**In vitro magnetic resonance imaging**. MRI data were acquired on a 7 T Bruker Biospec system using a $T_1$-weighted 2D gradient echo sequence (echo time, TE = 5

ms, repetition time, TR = 100 ms; flip angle, FA = 65°). Longitudinal relaxivity ($r_1$) measurements were acquired using a 2D spin echo sequence (TE = 11 ms, TR = 125, 200, 400, 800, 1500, 3000, and 5000 ms), with in-plane resolution of $200 \times 200$ μm$^2$ and 2 mm slice thickness. $R_1$ maps and values were generated using Karl Schmidt's MRI Analysis Calculator Plugin for ImageJ (National Institutes of Health, Bethesda, MD) or MATLAB scripts (Mathworks, Natick, MA) written in house. Stimulus-dependent $R_1$ changes in cells were calculated as $\Delta R_1/R_1 = [R_1(\text{incubated cells}) - R_1(\text{naive cells})]/R_1(\text{naive cells})$. Statistical comparisons between paired conditions were performed using Student's $t$-test, and all error bars denote the standard error of the mean (SEM) from three or more measurements, unless otherwise noted.

**In vitro characterization of ManICS1 and ManICS1-AM**. ManICS1, calcium and magnesium chloride stock solutions were all prepared in 25 mM 3-($N$-morpholino) propanesulfonic acid (MOPS), pH 7.4, with 100 mM KCl. Titration series were prepared as separate triplicates for each data point in the presence of constant concentrations of ManICS1, confirmed subsequently by inductively coupled plasma optical emission spectroscopy (ICP-OES). Samples were arrayed into microtiter plates and measured by MRI using a Bruker (Billerica, MA) Avance 7 T scanner. Unused wells were filled with buffer, and imaging was performed on a 2-mm slice through the sample. $K_d$ values were calculated by fitting a standard ligand-depleting binding equation to $R_1$ data. Error represents the standard error values for fitted parameters in a least squares fit to the data. Calcium affinity of manganese-free ManICS1 (10 μM) was determined by fitting a nondepleting binding model to spectroscopic absorbance data obtained at 230 nm, in the presence of buffered free calcium concentrations ranging from 0 to 1.5 μM. Titration data are shown with least squares curve fits. To measure stability against transmetallation, ManICS1-AM (40 μM) in Bis-Tris (50% EtOH, pH 7.4, 50 mM) was treated with solutions of CuCl$_2$ (50 μM) and ZnCl$_2$ (50 μM) in Bis-Tris (50% EtOH, pH 7.4, 50 mM) and stirred at room temperature for 5 h before recording of optical absorbance spectra.

**Cell labeling with ManICS1-AM**. HEK293 cells (Freestyle 293-F, Thermo Fisher Scientific, Waltham, MA) were cultured and prepared for relaxometry using a previously established method[19]. To assess uptake, cells were exposed to contrast agents in media for 30 min or 2 h then centrifuged at $750 \times g$ for 5 min, washed with Hank's buffered saline solution and repelleted. At this point, for time course experiments, washed cells were incubated in media again for varying time intervals and centrifuged again. In the final stage of preparation, cells were resuspended in media at $10^7$ cells per 100 μl, and at this point drugs were added for pharmacological stimulation experiments. The suspended cells were finally plated into wells of a 384-well plate and then immediately pelleted by 1 min centrifugation at $750 \times g$ for imaging.

Subcellular fractionation analysis of ManICS1-AM-labeled cells was conducted by addition of 0.05% saponin and pelleting the cells at $750 \times g$ to recover cytosolic fraction, resuspending the cells in low salt buffer and lysing the cells with 50 passes through a dounce homogenizer and a final centrifugation at $10,000 \times g$ with the supernatant collected as the nucleosolic/organellular fraction and the pellet collected as the membrane fraction. Each cell fraction was aliquoted into a crucible, incubated at 70 °C overnight to dessicate, then at 250 °C overnight to incinerate all organic molecules. The residual ash was resuspended in 200 μl 70% nitric acid and incubated at 70 °C overnight to soublize the metal oxides. The final residue was suspended into 2% nitric acid for analysis with ICP-OES.

**Cellular toxicity and viability assays**. HEK293 cells were incubated in 100 μM ManICS1-AM in media containing 5% DMSO for 30 min and then washed in media. Control cells were either untreated (naive), treated with saponin, or treated with DMSO vehicle only. To assess acute toxicity and membrane disruption, a subset of cells was incubated with 4 μM Ethidium Homodimer III (Biotium, Fremont, CA) and assayed for fluorescence at 530 and 620 nm. Higher fluorescence ratio (530/620) indicates increased cell penetrance and intercalation into DNA, indicative of toxicity. To assess long-term viability, an MTT assay (Life Technologies, Carlsbad, CA) was performed. Cells were incubated in the MTT reagent for 4 h at 37 °C to generate formazan crystals, which were then solubilized in sodium dodecylsulfate solution for 2 h at 37 °C and assayed for optical density at 570 nm. Higher absorption indicates higher NAD(P)H-dependent enzymatic activity, indicative of cell health.

**Measurement of intracellular calcium responses**. For stimulation experiments, cells were incubated with 10 μM ManICS1-AM for 2 h to allow for effective labeling and AM ester cleavage. Pharmacological stimulation was conducted by adding 5 μM thapsigargin, 25 μM carbochol, 5 μM calcimycin, or 30 μM arachidonic acid (Sigma-Aldrich, St. Louis, MO). Chemical agents were added while cells were in suspension to allow for mixing, then samples were gently pelleted the samples and collected $R_1$ relaxometry data.

For comparison stimulation measurements performed with Fura-2FF-AM (BioVision, Exton, PA), adherent HEK293 cells (Life Technologies) were seeded onto a 96-well plate at 5000 cells/well and grown for two days until 90% confluent. Cells were then incubated for 45 min in 5 μM Fura-2FF-AM and then washed with

media. Stimulants (10 µM thapsigargin, 100 µM charbocol, 5 µM calcimycin, or 30 µM arachidonic acid) were quickly added to multiple wells via multipipettor and fluorescence output was measured at 340 and 380 nm using a Spectramax plate reader. Measurements were repeated every 5 min for 40 min to chart the time course of calcium concentration changes at room temperature. Mean responses reported in Fig. 4d were reported as fold changes in the ratio of fluorescence emission at 340 and 380 nm pre- vs. post stimulation.

**Optogenetics.** BACCS2-IRES-Orai (Addgene 67627) was transiently transfected into 293freestyle cells using 293fectin (Life Technologies) and given 48 h to express. BACCS2-expressing and control cells were incubated in 40 µM ManICS1-AM for 2 h or 5 µM X-Rhodamine-1 (Life Technologies) for 30 min then washed with complete media without phenol red (Life Technologies). Cells were then pelleted cooled on ice and suspended in 0.5× volumes of Matrigel (Corning). The liquid Matrigel/cell slurry was then aliquoted onto a plate adapted to accommodate a fiber optic input, placing the cell directly in the lightpath but 2 mm from the fiber light source, to prevent possible heating artefacts. A 480-nm laser interfaced to the fiber was controlled via an Arduino microcontroller, using a custom written pulse sequence. Data were processed in ImageJ and analyzed using custom MATLAB scripts for high/low pass filtering, detrending using a polynomial fit to the pre-stimulus baseline and the last data point, with averaging across multiple experiments. Time course images were binned into 2 min averages centered around the peak response.

**Animal use.** Male Sprague-Dawley rats (250–300 g) were purchased from Charles River Laboratories (Wilmington, MA) and used for all in vivo experiments. Animals were housed and maintained on a 12 h light/dark cycle and permitted ad libitum access to food and water. All procedures were performed in strict compliance with US Federal guidelines, with oversight by the MIT Committee on Animal Care.

**In vivo validation experiments.** A 4-mM stock of ManICS-AM or control agent in DMSO was diluted to 200 mM in aCSF (125 mM NaCl, 2.5 mM KCl, 1.4 mM CaCl₂, 1 mM MgCl₂, 1.25 mM NaH₂PO₄, 25 mM NaHCO₃). Prior to intracranial infusion, animals were anesthetized with ~2% isoflurane, shaved, and mounted in a rodent stereotaxic device (David Kopf Instruments), while vital signs were monitored by pulse oximeter (Nonin Medical). The scalp over bregma was retracted and 28G holes were drilled into the skull 0.7 mm anterior and 3 mm lateral to bregma for access to the caudate nucleus based on a standard rat brain atlas[32]. 28G cannula guides that projected 2 mm below the surface of the skull, preassembled with 33G metal injection cannulae that projected 5 mm below the skull (Plastics One) were lowered to the appropriate depth through the holes and the guides were affixed in place using SEcure light curing dental cement (Parkell, Inc.) Intra-cranial infusions were performed using an injection rate of 0.12 µl min⁻¹ for 2 h with control vs. test sides alternated for each experiment to negate potential artifacts or bias. Metal cannulae were then removed and the animal was transferred to a custom MRI rat imaging cradle. 33G plastic cannulae (Plastics One) loaded with aCSF or stimulatory agent were then lowered bilaterally through the guide cannulae to the contrast agent delivery sites, and fixed in place with a locking cap (Plastics One). Neural stimulation was induced with isotonically KCl-supplemented aCSF (125 mM KCl, 2.5 mM NaCl, 1.4 mM CaCl₂, 1 mM MgCl₂, 1.25 mM NaH₂PO₄, 25 mM NaHCO₃), infused at a rate of 0.2 µl min⁻¹ for 5 min.

In vivo imaging experiments were conducted on a 9.4 T Biospec MRI scanner (Bruker) using a cross coil volume transmitter and surface receiver configuration. Functional imaging scans were acquired using a $T_1$-weighted gradient echo sequence (TE = 5.5 ms, TR = 150 ms, FA = 45°). These images had 48 × 32 matrix size with 400 µm in-plane resolution and 1 mm slice thickness, and were acquired with an acquisition rate of 4.6 s/image. Each experiment consisted of a 5 min pre-stimulation period, followed by 5 min of stimulation and 10 min of rest. Anatomical images used in Fig. 5 were obtained using a $T_1$-weighted rapid acquisition with refocused echoes (RARE) pulse sequence (TE = 8 ms, TR = 300 ms, FA = 160°, RARE factor = 4, number of averages = 20), with an in-plane resolution of 100 µm and a matrix size of 192 × 128 per slice.

Distribution of agent was determined by identifying the cannula tip and defining all contiguous pixels within 10% of image intensity at the tip as detected agent. The mean radius was determined for each resulting region of interest (ROI) and the volume occupied by the contrast agent was estimated as the volume of a sphere with this radius. Mean MRI signal within these ROIs was compared at multiple time points for analysis of contrast loss over time. The average tissue ManICS concentration was estimated from the contrast agent-dependent percent signal increase observed in $T_1$-weighted anatomical scans, using the fact that $T_1$-weighted MRI signal is proportional to $1 - \exp\{-R_1 \cdot TR\}$; $R_1$ in the absence of ManICS1 is taken as 0.31 s⁻¹ [33] and the $\Delta R_1$ due to contrast agent addition is the effective [ManICS1] times its in vitro calcium-free $r_1 = 3.6$ mM⁻¹ s⁻¹.

For analysis of dynamic signal changes during stimulation, MRI intensity was integrated over ROIs defined by a 1.2 × 1.2 mm square area adjacent to the cannula tip and centered on the point of peak signal change during stimulation period. Signal traces were filtered, normalized, and averaged using the same algorithm used for analysis of the optogenetic experiments. Percent signal changes were computed with respect to a baseline defined by the first 5 min of imaging and the last time

point. Mean signal change amplitudes reported in Fig. 5e were defined by the average signal change during the final 1 min of infusion. The map of percent signal change in Fig. 5d was generated by averaging percent signal change maps from five animals aligned to their respective cannula tips and plotting the result overlaid on a representative anatomical image. Sample sizes ($n \geq 5$) used for each in vivo functional experiment were appropriate for providing a statistical power of 0.8 for detection of a ≥4% signal change over a standard deviation of 2%.

**Code availability.** Scripts used for data analysis are available upon reasonable request.

**Reporting Summary.** Further information on experimental design is available in the Nature Research Reporting Summary linked to this article.

## Data availability
Raw MRI datasets generated during and/or analyzed during the current study are available from the corresponding author on reasonable request.

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

## Acknowledgements

Project funding was provided by NIH grants R01-DA038642, R21-MH102470, BRAIN Initiative awards U01-NS090451, U01-NS103470, and UF1-NS107712, and an MIT Simons Center for the Social Brain Seed Grant to A.J., as well as R01-GM065519 to S.J.L. We acknowledge the Department of Chemistry Instrumentation Facility for use of analytical equipment and also thank Miyeko D. Mana for assistance with the 3D gel experiments and Souparno Ghosh for additional support with tissue culture experiments.

## Author contributions

A.B. designed and conducted all synthesis, characterization, and in vitro experiments with assistance from C.G.W. B.B.B. designed and performed all cellular and in vivo experiments with assistance from C.G.W. and E.S.L. A.J., A.B., and B.B.B. designed the research and wrote the manuscript. S.J.L. advised on molecular design and synthetic procedures.

## Additional information

**Competing interests:** The authors declare no competing interests.

