## [Peer Review File · Nature Communications]

Reviewers' comments:

Reviewer #1 (Remarks to the Author):

Barandov et al describe the use of manganese-based paramagnetic contrast agent that allows for non invasive monitoring of intracellular calcium signaling in vivo. The authors compellingly demonstrate that MR signal can be modulated by pharmacological and optogenetic tools both in vitro and in vivo, and show that the recorded responses parallel signal obtained with fluorescent calcium indicators.

The manuscript is very well written and describes a new tool for non invasive molecular imaging that could have a wide impact in the field. The design of the sensor and its in vitro validation are described rigorously, elegantly, and in great detail.

The only notable shortcoming of the manuscript in its current form is the lack of a more extensive in vivo validation of this tool, which limits the breadth of the article, and prevents a fair assessment of the actual in vivo utility of this cleverly-designed and highly promising approach. I believe the work would have a broader impact and if the authors could follow up their initial validation with additional physiologically-relevant stimuli targeting specific brain regions in vivo. This can be addressed easily for example by assessing the sensitivity of this tool to sensory stimulation (e.g. forepaw stimulations), or tonic optogenetic stimulations of excitatory neurons, or the administration of excitatory psychostimulant drugs that alter intracellular calcium signalling and regional brain activity. I think the addition of one or more of such datasets is warranted to obtain a more robust assessment of the in vivo potential of this tool as a complement to existing calcium sensing approaches.

Reviewer #2 (Remarks to the Author):

This paper presents a novel MRI contrast agent that has Ca²⁺ dependent relaxation properties with potential for allowing in vivo Ca imaging of deep tissue. Although this is still a small study demonstrating the in vivo capabilities, the experimental lead up to that with studies in solution and in cells are well thought out and documented. Of the potential candidate issues of conjugate stability, membrane transport, Ca²⁺ specificity and the deltaR1 sensitivity are all addressed for the proposed agent and using appropriate MRI methodology. The in vivo assessments use direct injection of the agent into rat brain and significant T1 changes are observed in relation to altering Ca⁺ via CSF infusion. In future expts and for the agent to have a general applicability, I would imagine it will be necessary to address the issue of getting the agent across the blood brain barrier, maybe some added discussion on practical issues to further develop this agent could be added or whether there are likely limitations in how it is used for neuro work. also no mention is made of neurotoxicity. This agent was developed from optical imaging agents - so is the toxicity profile expected to be the same? There is demonstrated some small concentration dependent effects on relaxivity in supplementary Fig 3. Although this was dismissed as insignificant maybe the authors can comment further in relation to the actual intracellular concentrations of the agent that are expected to occur for in vivo expts. SUp Fig 3 would suggest that at low Ca²⁺ a gradual accumulation of the agent would lead to reduced r1 (and so reduced T1w image intensity with time), whereas at high Ca²⁺ the r1 would increase with increased agent accumulation. Would these be significant effects in time course experiments. But these are just minor comments on a very nice piece of work with good potential for expanding the capability of experimental in vivo MRI.

Reviewer #3 (Remarks to the Author):

Jasanoff et al reports the first example of MRI monitoring of intracellular Ca concentration changes

in vitro in cells as well as in a living rat brain. The quest for molecular imaging approaches to monitor brain activity started more than 10 years ago and this report is a very important step in the field of neuroimaging. The same group succeeded in detecting in vivo neurotransmitter concentration changes by using paramagnetic engineered proteins, here again they provide a beautiful demonstration of the power of molecular MR imaging.

The chemical design of their Ca-sensitive MRI agent is based on elements which were previously validated individually, such as the BAPTA-type Ca chelator and a Mn³⁺ complex as an MRI reporter. The BAPTA chelator is used in an ester form to facilitate cellular internalization, followed by enzymatic cleavage of the esters to provide the final Ca-binding unit. The in vitro step-by-step validation of the combined Ca-binding and Mn-containing molecule follows a very logical study design and leads to a comprehensive characterization. Although the thermodynamic stability of the Mn(III) complex was partially assessed in a previous JACS publication, it would seem interesting to conduct a similar type of spectrophotometric experiment as in Fig. 1d in the presence of physiological Zn and Cu concentrations to exclude potential Zn transmetallation.

The imaging experiments in cells and in vivo are carried out with the highest proficiency and constitute a clear evidence of the potential of this novel agent for Ca detection. I strongly believe that this work will influence thinking in the field.

Before publication, the authors should consider the following points:

The expected concentration variations should be indicated in l. 156.

Lines 214-215: how to rationalize the difference in the wash out of the two compounds which are in fact the same after enzymatic cleavage of the esters?

l. 230: why should protein binding influence the Ca binding affinity, especially in this sense?

We thank the reviewers for their thoughtful comments on our manuscript. Responses are supplied below, along with cross references to the text where appropriate. Significant changes are also highlighted in red in an attached version of the revised manuscript and supplement that also includes two new supplementary figures.

Reviewer 1

Barandov et al describe the use of manganese-based paramagnetic contrast agent that allows for noninvasive monitoring of intracellular calcium signaling in vivo. The authors compellingly demonstrate that MR signal can be modulated by pharmacological and optogenetic tools both in vitro and in vivo, and show that the recorded responses parallel signal obtained with fluorescent calcium indicators. The manuscript is very well written and describes a new tool for noninvasive molecular imaging that could have a wide impact in the field. The design of the sensor and its in vitro validation are described rigorously, elegantly, and in great detail.

We are grateful for this assessment.

The only notable shortcoming of the manuscript in its current form is the lack of a more extensive in vivo validation of this tool, which limits the breadth of the article, and prevents a fair assessment of the actual in vivo utility of this cleverly-designed and highly promising approach. I believe the work would have a broader impact and if the authors could follow up their initial validation with additional physiologically-relevant stimuli targeting specific brain regions in vivo. This can be addressed easily for example by assessing the sensitivity of this tool to sensory stimulation (e.g. forepaw stimulations), or tonic optogenetic stimulations of excitatory neurons, or the administration of excitatory psychostimulant drugs that alter intracellular calcium signalling and regional brain activity. I think the addition of one or more of such datasets is warranted to obtain a more robust assessment of the in vivo potential of this tool as a complement to existing calcium sensing approaches.

We are in complete agreement with the Reviewer on the need for further *in vivo* validation as the technology we present here develops. On the other hand, we have designed the current manuscript primarily for presenting the chemistry and initial validation of our new tool, and expect to present more involved *in vivo* testing and discovery-oriented applications in future publications. Importantly, this allows us to maintain equal contribution status between the chemist and the biologist who now serve as co-first authors. In addition, we feel that the scope and length of the manuscript are already well-matched to the standards of *Nature Communications*, without further *in vivo* results.

Reviewer 2

This paper presents a novel MRI contrast agent that has Ca²⁺ dependent relaxation properties with potential for allowing in vivo Ca imaging of deep tissue. Although this is still a small study demonstrating the in vivo capabilities, the experimental lead up to that

with studies in solution and in cells are well thought out and documented. Of the potential candidates issues of conjugate stability, membrane transport, Ca²⁺ specificity and the deltaR1 sensitivity are all addressed for the proposed agent and using appropriate MRI methodology. The in vivo assessments use direct injection of the agent into rat brain and significant T1 changes are observed in relation to altering Ca⁺ via CSF infusion.

Thank you for noting these experiments.

In future expts and for the agent to have a general applicability, I would imagine it will be necessary to address the issue of getting the agent across the blood brain barrier, maybe some added discussion on practical issues to further develop this agent could be added or whether there are likely limitations in how it is used for neuro work.

Yes, for general applicability in neuroimaging, the agent will have to be delivered in some way past the blood-brain barrier. The current agent is designed for membrane permeability and as such may display spontaneous BBB permeability, but there could be a need for further steps to improve brain localization beyond this. In the revised manuscript, we have added text to p. 13 to address this point.

also no mention is made of neurotoxicity. This agent was developed from optical imaging agents - so is the toxicity profile expected to be the same?

Like optical calcium agents, ManICS1 has the potential to perturb calcium levels in cells, but this is not expected to be problematic except at very high probe concentrations. To directly address issues of toxicity however, we have now performed assays using two separate tests for acute toxicity and long-term viability in cell culture. The results show no significant effect of ManICS1 compared with vehicle in either assay. Data are presented in the new Supplementary Fig. 7 and referred to on pp. 10 and 16-17 of the revised text.

There is demonstrated some small concentration dependent effects on relaxivity in supplementary Fig 3. Although this was dismissed as insignificant maybe the authors can comment further in relation to the actual intracellular concentrations of the agent that are expected to occur for in vivo expts.

From the mean signal change of ~20% upon ManICS1 injection, as reported in Fig. 4b, and the relaxivity of the contrast agent in the absence of calcium ($3.6 \text{ mM}^{-1}\text{s}^{-1}$), we can use the T_1 -dependence of MRI signal proportional to $(1 - \exp\{-R_1TR\})$, with a TR value of 0.4 s and measured basal R_1 of 0.31 s^{-1} in rat brain at 9.4 T, to estimate a mean *in vivo* concentration of about 19 μM in tissue. The intracellular volume fraction is around 80% in the brain, so if the contrast agent is entirely intracellular, it would have an average concentration of 24 μM . Our data on the concentration dependence of ManICS1 relaxivity suggest that its relaxivity is relatively concentration independent until much higher concentrations are reached. We now comment on this point on pp. 12 and 20 in the revised manuscript.

Supp Fig 3 would suggest that at low Ca²⁺ a gradual accumulation of the agent would lead to reduced r₁ (and so reduced T_{1w} image intensity with time), whereas at high Ca²⁺ the r₁ would increase with increased agent accumulation. Would these be significant effects in time course experiments.

The horizontal axis in Supplementary Fig. 3 reports total calcium levels including both unbound and bound calcium ions. The decreasing effective r_1 for higher concentrations of agent at low $[Ca^{2+}]$ thus reflects the lower saturation fraction of ManICS1 under conditions where the stoichiometric ratio of contrast agent to calcium is high. For any concentration of ManICS1 (c) and buffered free calcium concentration ($[Ca^{2+}]$), the change in R_1 from baseline conditions in the absence of calcium would be $\Delta R_1 = c\Delta r_1[Ca^{2+}]/(K_d + [Ca^{2+}])$, where Δr_1 is the calcium-dependent change in ManICS1 relaxivity and K_d is the apparent dissociation constant of ManICS1 for calcium. More agent at a given unbound $[Ca^{2+}]$ would always lead to greater ΔR_1 and greater T_1 -weighted signal. Although changes in contrast agent concentration could nevertheless act as confounds in calcium-dependent imaging, in general such changes would be slow compared with calcium concentration fluctuations associated with cell signaling. We have amended Supplementary Fig. 3 and its caption to help clarify these points.

But these are just minor comments on a very nice piece of work with good potential for expanding the capability of experimental in vivo MRI.

We thank the reviewer for this assessment.

Reviewer 3

Jasanoff et al reports the first example of MRI monitoring of intracellular Ca concentration changes in vitro in cells as well as in a living rat brain. The quest for molecular imaging approaches to monitor brain activity started more than 10 years ago and this report is a very important step in the field of neuroimaging. The same group succeeded in detecting in vivo neurotransmitter concentration changes by using paramagnetic engineered proteins, here again they provide a beautiful demonstration of the power of molecular MR imaging. The chemical design of their Ca-sensitive MRI agent is based on elements which were previously validated individually, such as the BAPTA-type Ca chelator and a Mn³⁺ complex as an MRI reporter. The BAPTA chelator is used in an ester form to facilitate cellular internalization, followed by enzymatic cleavage of the esters to provide the final Ca-binding unit. The in vitro step-by-step validation of the combined Ca-binding and Mn-containing molecule follows a very logical study design and leads to a comprehensive characterization.

We are grateful for this evaluation.

Although the thermodynamic stability of the Mn(III) complex was partially assessed in a previous JACS publication, it would seem interesting to conduct a similar type of spec-

trophotometric experiment as in Fig. 1d in the presence of physiological Zn and Cu concentrations to exclude potential Zn transmetallation.

We have addressed this important issue with a new Supplementary Figure 5 and additional text on p. 8 of the revised manuscript. Transmetallation of the agent by 50 μM Zn^{2+} or Cu^{2+} was negligible on a time scale of hours.

The imaging experiments in cells and in vivo are carried out with the highest proficiency and constitute a clear evidence of the potential of this novel agent for Ca detection. I strongly believe that this work will influence thinking in the field.

We thank the Reviewer for noting this.

*Before publication, the authors should consider the following points:
The expected concentration variations should be indicated in l. 156.*

This is now indicated on p. 7.

Lines 214-215: how to rationalize the difference in the wash out of the two compounds which are in fact the same after enzymatic cleavage of the esters?

The washout data follow preincubation of cells with ManICS1 and ManICS1-AM. During the preincubation, the esterified version should have penetrated cells to a greater extent than the non-esterified version, and become trapped there upon cleavage of its esterase groups. Thus, the slower washout of contrast from the cells incubated with ManICS1-AM results from intracellular compartmentalization of the polar agent. We have added text on p. 9 of the revised manuscript to clarify this point.

l. 230: why should protein binding influence the Ca binding affinity, especially in this sense?

Protein binding can alter the conformational equilibrium of ManICS1, altering its propensity to experience intramolecular interactions between the BAPTA and manganese chelating domains. Because this equilibrium is coupled to calcium binding, the calcium binding affinity would be affected. Coordination interactions could also be affected by transient binding between ManICS1 and macromolecules. For instance, if a protein coordinated one of the axial manganese sites, the affinity of the other axial site for intramolecular BAPTA binding could well be altered. We now briefly comment on this subject on p. 10 of the revised text.

REVIEWERS' COMMENTS:

Reviewer #2 (Remarks to the Author):

I thank the authors for clearly addressing the points in my review. I have no further comments, and congratulate the authors again on their work.

Reviewer #3 (Remarks to the Author):

The different points raised in the previous report have been addressed appropriately. The manuscript can be accepted in the present form, without further corrections.